# Face Presentation Attack Detection Using Deep Background Subtraction

**DOI:** 10.3390/s22103760

**Published:** 2022-05-15

**Authors:** Azeddine Benlamoudi, Salah Eddine Bekhouche, Maarouf Korichi, Khaled Bensid, Abdeldjalil Ouahabi, Abdenour Hadid, Abdelmalik Taleb-Ahmed

**Affiliations:** 1Laboratoire de Génie Électrique, Faculté des Nouvelles Technologies de l’Information et de la Communication, Université Kasdi Merbah Ouargla, Ouargla 30 000, Algeria; benlamoudi.azeddine@univ-ouargla.dz (A.B.); korichi.maarouf@univ-ouargla.dz (M.K.); bensid.khaled@univ-ouargla.dz (K.B.); 2Department of Computer Science and Artificial Intelligence, Faculty of Informatics, University of the Basque Country UPV/EHU, 20018 San Sebastian, Spain; sbekhouche001@ikasle.ehu.eus; 3UMR 1253, iBrain, INSERM, Université de Tours, 37000 Tours, France; 4Institut d’Electronique de Microélectronique et de Nanotechnologie (IEMN), UMR 8520, Université Polytechnique Hauts de France, Université de Lille, CNRS, 59313 Valenciennes, France; abdenour.hadid@ieee.org (A.H.); abdelmalik.taleb-ahmed@uphf.fr (A.T.-A.)

**Keywords:** biometrics, face presentation attack, deep learning

## Abstract

Currently, face recognition technology is the most widely used method for verifying an individual’s identity. Nevertheless, it has increased in popularity, raising concerns about face presentation attacks, in which a photo or video of an authorized person’s face is used to obtain access to services. Based on a combination of background subtraction (BS) and convolutional neural network(s) (CNN), as well as an ensemble of classifiers, we propose an efficient and more robust face presentation attack detection algorithm. This algorithm includes a fully connected (FC) classifier with a majority vote (MV) algorithm, which uses different face presentation attack instruments (e.g., printed photo and replayed video). By including a majority vote to determine whether the input video is genuine or not, the proposed method significantly enhances the performance of the face anti-spoofing (FAS) system. For evaluation, we considered the MSU MFSD, REPLAY-ATTACK, and CASIA-FASD databases. The obtained results are very interesting and are much better than those obtained by state-of-the-art methods. For instance, on the REPLAY-ATTACK database, we were able to attain a half-total error rate (HTER) of 0.62% and an equal error rate (EER) of 0.58%. We attained an EER of 0% on both the CASIA-FASD and the MSU MFSD databases.

## 1. Introduction

Individuals can be successfully identified and authenticated using biometric features and traits. Hence, it is appropriate for access control and global security systems that depend on person recognition, which is achieved through the use of a variety of biometric modalities, ranging from the classic fingerprint through the face, iris, ear [1,2,3,4] and, more recently, vein and blood flow. Furthermore, a number of spoofing methods have been developed in order to overcome such biometric systems [5]. When someone tries to get around a face biometric system by placing a fake face in front of the camera, this is known as a presentation attack. Nevertheless, compared to other modalities, the abundance of still face images or video sequences on the internet has made it exceptionally easy to obtain a person’s facial data.

The spoofing detection literature discusses multiple types of presentation attack instruments, such as print, replay, silicon masks, and makeup attacks. The focus of our work is on the first two attacks, namely print and replay attacks. The print attack spoofs 2D face recognition systems by using printed photographs of a subject, whereas the replay attack presents a video of a bona fide presentation to avoid liveness detection. Furthermore, the low cost of launching a face presentation attack instrument has increased the prevalence of the problem. Face recognition system spoofing media ranges from low-quality paper prints to high-quality photographs, as well as video streams played in front of the biometric authentication system sensor.

Feature extraction is a critical component of the face presentation attack detection task when using a classical machine learning classifier. Convolutional neural network(s) (CNN) can also be used to predict scores. This latter is a crucial component of deep learning algorithms, such as the ResNet-50 [6] pre-trained model, which has been studied for a few years under a variety of conditions and scenarios. In our work, we used background subtraction (BS) with CNN to predict each frame in the input video and rank the score using the MV algorithm to determine whether the input video is real or fake.

Inspired by the work of frame difference and multilevel representation (FDML) [7], we propose an effective system for face presentation attack detection. To do this, we suggest using the background substruction method in the preprocessing step to adjust the face’s motion. The MV algorithm is used to improve the performance rate as well as the decision of the input video after predicting the score of each frame by ResNet-50. To test our system, we used videos from numerous public face spoof databases with varying quality, resolutions, and dynamic ranges. We also compared our results to those of a number of current state-of-the-art approaches. The following are the main contributions of this work:Improving face presentation attack detection using BS that discriminates the motion of real face from a fake one.Fine-tuning the ResNet-50 model for the face presentation attack detection task to extract meaningful deep facial features.Using the MV algorithm to increase the classification rate of the system, which is clearly observed when the methodology outperformed previous methodologies in the literature, according to the results of our experiments.Tackling the sensor interoperability problem by including the experiments of inter-database and intra-database tests.

The remainder of the paper is structured as follows. Section 2 describes related work on face presentation attack detection. Then, our approach is described in detail in Section 3. Section 4 summarizes the experimental results and provides a comparative analysis. The section also describes the databases that we used in our tests. Section 5 draws some conclusions and highlights some future directions. Abbreviations define the main acronyms used in this paper.

## 2. Related Work

Presentation attack can be detected in a variety of ways. In this paper, we focus on two types of face presentation attack detection methods: handcrafted and deep learning-based methods. In this section, we present most previous work in face presentation attack detection. However, we only focus on those that are thematically closer to our goals and contributions.

### 2.1. Handcraft-Based Techniques

Texture features, which can describe the contents and details of a specific region in an image, are an important low-level feature in face presentation attack detection methods. Therefore, the analysis of image texture information is used in many techniques, such as compressed sensing, which preserves texture information and denoising at the same time [8,9]. These techniques based on handcrafted features provide accurate features that increase the detection rate of a spoofing system. Smith et al. [10] proposed a method for countering attacks on face recognition systems by using the color reflected from the user’s face as displayed on mobile devices. The presence or absence of these reflections can be utilized to establish whether or not the images were captured in real time. The algorithms use simple RGB images to detect presentation attack. These strategies can be classified into two categories: static and dynamic approaches. The static is used on a single image, whilst dynamic is used on the video.

The majority of approaches for distinguishing between real and synthetic faces are focused on texture analysis. Arashloo et al. [11] combined two spatial–temporal descriptors using kernel discriminant analysis fusion. They are multiscale binarized statistical image features on three orthogonal planes (MBSIF-TOP) and multiscale local phase quantization on three orthogonal planes (MLPQ-TOP). To distinguish between real and fake individuals, Pereira et al. [12] also experimented with a dynamic texture that was based on local binary pattern on three orthogonal planes (LBP-TOP). The good results of LBP-TOP are due to the fact that temporal information is crucial in face presentation attack detection. Tirunagari et al. [13] used local binary pattern(s) (LBP) for dynamic patterns and dynamic mode decomposition (DMD) for visual dynamics. Wen et al. [14] proposed an image distortion analysis-based method (IDA). To represent the face images, four different features were used: blurriness, color diversity, specular reflection, and chromatic moments, also relying on the features that can detect differences between real image and fake one without capturing any information about the user’s identity. Patel et al. [15] investigated the impact of different RGB color channels (R, G, B, and gray scale) and different facial regions on the performance of LBP and dense scale invariant feature transform (DSIFT) based algorithms. Their investigations revealed that extracting the texture from the red channel produces the best results. Boulkenafet et al. [16] proposed a color texture analysis-based face presentation attack detection approach. They employed the LBP descriptor to extract texture features from each channel after encoding the RGB images in two color spaces: HSV and YCbCr, and then concatenated these features to distinguish between real and fake faces.

Some methods, such as [17], have recently used user-specific information to improve the performance of texture-based FAS techniques. Garcia et al. [18] proposed face presentation attack detection by looking for Moiré patterns caused by digital grid overlap where their detection is based on frequency domain peak detection. For classification, they used support vector machines (SVM) with an radial basis function kernel. They started to run their tests on the Replay Attack Corpus and Moiré databases. Other face presentation attack detection solutions are based on textures on 3D models, such as those used in [19]. Because the attacker in 3D models utilizes a mask to spoof the system, the introduction of wrinkles might be extremely helpful in detecting the attack. The presented work in [19] examines the viability of performing low-cost assaults on 2.5D and 3D face recognition systems using self-manufactured three-dimensional (3D) printed models.

### 2.2. Deep Learning-Based Techniques

Actually, deep learning is used in a variety of systems and applications for biometric authentication [20], where the deep network can be trained using a number of patterns. After learning all of the dataset’s unique features, the network can be used to identify similar patterns. Deep learning approaches have mostly been used to learn face presentation attack detection features. Moreover, deep learning is efficient at classification (supervised learning) and clustering tasks (unsupervised learning). Thus, the system assigns class labels to the input instances in a classification task, but the instances in clustering approaches are clustered based on their similarity without the usage of class labels.

To train models with significant discriminative abilities, Yang et al. [21] used a deep CNN rather than manually constructing features from scratch. Quan et al. proposed a semi-supervised learning-based architecture to fight face presentation attack threats using only a few tagged data, rather than depending on time-consuming data annotations. They assess the reliability of selected data pseudo labels using a temporal consistency requirement. As a result, network training is substantially facilitated. Moreover, by progressively increasing the contribution of unlabeled target domain data to the training data, an adaptive transfer mechanism can be implemented to eliminate domain bias. According to the authors in [22], they use a type of ground through (GT) termed appr-GT in conjunction with the identity information of the spoof image to generate a genuine image of the appropriate subject in the training set. A metric learning module constrains the generated genuine images from the spoof images to be near the appr-GT and far from the input images. This reduces the effect of changes in the imaging environment on the appr-GT and GT of a spoof image.

Jia et al. [23] proposed a unified unsupervised and semi-supervised domain adaptation network (USDAN) for cross-scenario face presentation attack detection, with the purpose of reducing the distribution mismatch between the source and target domains. The marginal distribution alignment module (MDA) and the conditional distribution alignment module (CDA) are two modules that use adversarial learning to find a domain-invariant feature space and condense features of the same class.

Raw optical flow data from the clipped face region and the complete scene were used to train a neural network by Feng’s team et al. [24]. Motion-based presentation attack detection does not need a scenic model or motion assumption to generalize. They present an image quality-based and motion-based liveness framework that can be fused together using a hierarchical neural network.

In their work [25], Liu et al. proposed a deep tree network (DTN) that learns characteristics in a hierarchical form and may detect unanticipated presentation attack instrument by identifying the features that are learned.

Yu et al. [26] introduces two new convolution and pooling operators for encoding fine-grained invariant information: central difference convolution (CDC) and central difference pooling (CDP). CDC outperforms vanilla convolution in extracting intrinsic spoofing patterns in a number of situations.

As described in Qin et al. [27], adaptive inner-update (AIU) is a novel meta learning approach that uses a meta-learner to train on zero- and few-shot FAS tasks utilizing a newly constructed Adaptive Inner update Meta Face Anti spoofing (AIM-FAS).

According to Yu et al. [28], the multi-level feature refinement module (MFRM) and material-based multi-head supervision can help increase BCN’s performance. In the first approach, local neighborhood weights are reassembled to create multi-scale features, while in the second, the network is forced to acquire strong shared features in order to perform tasks with multiple heads.

CDC-based frame level FAS approaches, proposed by the authors in [29], have been developed. These patterns can be captured by aggregating information about intensity and gradient. In comparison to a vanilla convolutional network, the central difference convolutional network (CDCN) built with CDC has a more robust modeling capability. CDCN++ is an improved version of CDCN that incorporates the search backbone network with the multiscale attention fusion module (MAFM) for collecting multi-level CDC features effectively.

Spatiotemporal anti-spoof network (STASN) is a new attention mechanism invented by Yang et al. [30] that combines global temporal and local spatial information, allowing them to examine the model’s understandable behaviors.

To improve CNN generalization, Liu et al. [31] proposed to use innovative auxiliary information to supervise CNN training. A new CNN-RNN architecture for learning the depth map and rPPG signal from end-to-end is also proposed.

Wang et al. [32] proposed a depth-supervised architecture that can efficiently encode spatiotemporal information for presentation attack detection and develops a new approach for estimating depth information from several RGB frames. Short-term extraction is accomplished through the use of two unique modules: the optical flow-guided feature block (OFFB) and the convolution gated recurrent units (ConvGRU). Jourabloo et al. [33] proposed a new CNN architecture for face presentation attack detection, with appropriate constraints and supplementary supervisions, to discern between living and fake faces, as well as long-term motion. In order to detect presentation attacks effectively and efficiently, Kim et al. [34] introduced the bipartite auxiliary supervision network (BASN), an architecture that learns to extract and aggregate auxiliary information.

Huszár et al. [35] proposed a deep learning (DL) approach to address the problem of presentation attack instruments occurring from video. The approach was tested in a new database made up of several videos of users juggling a football. Their algorithm is capable of running in parallel with the human activity recognition (HAR) in real-time. Roy et al. [36] proposed an approach called the bi-directional feature pyramid network (BiFPN) to detect presentation attacks because the approach containing high-level information demonstrates negligible improvements. Ali et al. [37]—based on stimulating eye movements by using the use of visual stimuli with randomized trajectories to detect presentation attack instrument. Ali et al. [38]—by the combination of two methods, which are head-detection algorithm and deep neural network-based classifiers. The test involved various face presentation attacks in thermal infrared in various conditions.

It appears that most of the existing handcraft and deep learning-based features may not be optimal for the FAS task due to the limited representation capacity for intrinsic spoofing features. In order to learn more robust features for the domain shift as well as more discriminative patterns for liveness detection, we propose deep background subtraction and majority vote algorithm to take into account both dynamic and static information.

## 3. Proposed Approach

Figure 1 describes the overall structure of our proposed approach, which is divided into three modules: background subtraction, feature learning, and data classification. To begin, we used the background subtraction between consecutive frames to extract motion, we can also call this technique BS. Then, the features were extracted using the ResNet-50 transfer learning model on the foreground of BS. Finally, to distinguish between real and fake faces of each frame, we used a classification layer that employed a fully connected layer. After that, we used MV to predict whether the input video was real or not. In the subsections that follow, all subsystems (modules) will be discussed.

### 3.1. Background Subtraction Module

In our research work, we used a face presentation attack detection system based on an extended BS algorithm. The background subtraction approach, which is based on the premise of obtaining the pixels in the image sequence difference operation to do two or three continuous frames, is the most commonly used action target detection measure. Using an image pixel value obtained by subtracting the difference image and the binarized difference image, if the pixel value change threshold is less than a predefined one, we can feel this as a background pixel in the adjacent frame. If the pixel value of an image area changes dramatically, it is possible to deduce that this is due to the action of detecting spoof in the image caused by these symbols as foreground pixel regions. While taking dynamic information into account, a pixel region based on symbolic actions can determine the position of the target in the image.

Background subtraction is applied to images and the threshold result is displayed as a foreground image. Figure 2 shows an example of the output. This is a low-cost and ineffective method of detecting motion in a video stream. The image $P_{t}$ is transformed into a grey-scale (intensity) image $I_{t}$. Then, given the image $I_{t}$ and the previous image $I_{t-1}$, the current output is $R_{t}$, where:(1)$$R_{t}\left(x,y\right)=\left\{\begin{array}{cc} I_{t}\left(x,y\right)\; & \;\mathrm{if}\;|I_{t}\left(x,y\right)-I_{t-1}\left(x,y\right)|>T\\ 0\; & \;\mathrm{otherwise} \end{array}\right.$$

*T* is the value of the threshold parameter. In our situation, we just utilized a threshold to remove the pixels with the same values across the two frames. The foreground pixels take the value of the current frame if there is motion. The foreground pixel is set to zero if there is no motion.

We apply a moving window mechanism that takes the first frame within a window of 5 frames. This means that we take a frame and drop 4 frames in each cycle. For example, in a video of 150 frames, we use 30 frames and drops 120 frames.

### 3.2. Feature Learning Module

Feature learning (FL) is a set of approaches in machine learning that allow a system to discover the representation needed for feature detection, prediction, or classification from a preprocessed dataset automatically. This enables a machine to learn the features and apply them to a specific task-like classification and prediction. Feature learning can be achieved in deep learning by either creating a complete CNN to train and test the collection of images or adapting a pre-trained CNN for classification or prediction for the new images-set. Transfer learning is the latter strategy used in the deep learning domain. Transfer learning is a machine learning technique in which a model created for one task is utilized as the basis for a model on a different task.

Transfer learning (TL) is commonly used in deep learning (DL) applications to allow you to use a pre-trained network for solving new classification tasks. To meet the new learning tasks, the learning parameters of the pre-trained network with randomly initialized weights must be fine-tuned. Transfer learning is typically considerably faster and easier to learn/train than building a network from the initial concept. Transfer learning is an optimization and a quick way that can save time or improve efficiency.

In this section, a transfer learning technique is applied by fine-tuning a pretrained ResNet-50 model on ImageNet dataset using multiple spoofing datasets where the output of the last FC layer is changed to output two classes (real/fake). The network is called ResNet-50, due to the fact that it has 48 convolution layers, along with 1 MaxPool and 1 Average Pool layer, and it introduced the use of residual blocks.

### 3.3. Classification Module

Data classification is a vital process for separating large datasets into classes for decision-making, pattern detection, and other purposes. For multi-class classification problems with mutually exclusive classes, a classification layer uses a fully connected layer to compute the cross-entropy loss.

The features from ResNet-50 are passed via a FC layer made of 1024 neurons with a 40% dropout to prevent over-fitting in the classification module. Having followed that, the units were activated with a rectification mechanism called ReLU. MAX (X, 0) is the ReLu function, which sets all negative values in the matrix *X* to zero while keeping all other values constant. The reason for choosing ReLU is that deep network training with ReLU tended to converge considerably faster and more reliably than deep network training with sigmoid activation. Finally, the output layer comprised of one neuron unit programmed to compute probabilities for the classes using the sigmoid function (binary classifier). A sigmoid is a mathematical function that takes a vector of k real values and changes it to a two-probability probability distribution.

We employed voting ensemble in our tests to classify each subject (video) as real or fake. A voting ensemble (sometimes known as a "majority voting ensemble") is a machine learning model that incorporates predictions from several other models, such as multiple predictions in each frame after the input video’s last layer (classification layer). The predictions for each label are combined, and the label with the majority vote is forecasted (see Figure 1, classification module) to determine if the input video belongs to a real or fake one. The majority voting ensemble creates forecasts based on the most common one. It is a strategy that can be utilized to boost performance, with the goal of outperforming every frame used independently in the ensemble.

## 4. Experimental Results and Analysis

In this section, the employed benchmark datasets will be introduced first, followed by a brief description of the evaluation criteria. After that, we present and analyze a series of experiments that we assume demonstrate the efficacy of the proposed BS-CNN+MV based face presentation attack detection technique.

### 4.1. Database and Protocol

In order to assess of the effectiveness of our proposed presentation attack detection technique, we performed a set of experiments on well known databases where the top three challenging databases were used: the CASIA-FASD (http://www.cbsr.ia.ac.cn/english/FaceAntiSpoofDatabases.asp (accessed on 4 July 2014)); face anti-spoofing database, Replay-Attack (https://www.idiap.ch/dataset/replayattack (accessed on 6 August 2014)) database, and MSU (https://drive.google.com/drive/folders/1nJCPdJ7R67xOiklF1omkfz4yHeJwhQsz (accessed on 8 May 2014)) mobile face spoofing databases. Those databases contain video recordings of real and fake attacks. A brief description of these databases is given as fellow:

The CASIA-FASD database [39] is a dataset for face presentation attack detection. This database contains 50 genuine subjects in total and the corresponding fake faces are captured with high quality from the original ones. Therefore each subject contains 12 videos (3 genuine and 9 fake) under three different resolutions and light conditions, namely low quality, normal quality, and high quality. Moreover, three fake face attacks are designed, which include warped photo attack, cut photo attack, and video attack. The overall database contains 600 video clips and the subjects are divided into subsets for performing training and tests in which 240 videos of 20 subjects are used for training and 360 videos of 30 subjects for testing. Test protocol is provided, which consists of 7 scenarios for a thorough evaluation from all possible aspects (see Figure 3).

Among the popular databases designed for the presentation attack detection application, one can find the Replay-Attack database [40]. This database consists of 1300 video of real-access and attack attempts to 50 subjects, (See Figure 4). However, these videos were taken using a built-in webcam on a MacBook laptop under two separate scenarios (controlled and adverse). In addition, two cameras were used to create the faked facial attack for each person in high-resolution images and videos: a Canon PowerShot SX150 IS and an iPhone 3GS camera. Moreover, fixed attacks and hand attacks are the two types of attacks. There are ten videos in each subset: four mobile attacks with a resolution of 480×320 pixels on an iPhone 3GS screen, then, by using a first generation iPad with a screen resolution of 1024×768 pixels, four high-resolution screen attacks were performed. On A4 paper, two hard-copy print attacks (printed on a Triumph-Adler DCC 2520 colour laser printer) occupied the whole available printing surface. It should be noted that the complete set of videos is divided into three non-overlapping subsets for training, development, and testing in order to evaluate them.

The pattern recognition and image processing (PRIP) group at Michigan State University developed a publicly available MSU-MFSD [14] database for face presentation attacks. The database contains 280 video clips of attempted photo and video attacks on 35 clients. It was created using a mobile phone to capture both genuine and presentation attack. This was accomplished using two types of cameras: (1) the built-in camera in the MacBook Air 13 inch (640×480) and (2) the front-facing camera on the Google Nexus 5 Android phone (720×480). Each subject received two video recordings, the first of which was taken using a laptop camera and the second with an Android camera (See Figure 5). High-resolution video was recorded for each subject utilizing two devices to create the attacks: (1) Canon PowerShot 550D SLR camera, which captures 18.0 megapixel photos and 1080p high-definition video clips; (2) iPhone 5S back-facing camera, which captures 1080p video clips. There are three types of presentation attack instruments, the first one (1) high-resolution replay video, the first type of presentation attack instrument is a high-resolution replay video attack using an iPad Air screen, with a resolution of 2048×1536, the second is a mobile phone replay video attack using an iPhone 5S screen, with a resolution of 1136×640, and the third is a printed photo attack using an A3 paper with a fully-occupied printed photo of the client’s biometry, with a paper size of: 11×17 (279 mm× 432 mm), printed with an HP LaserJet CP6015xh printer at a resolution of 1200 × 600 dpi. Finally, to assess performance, the 35 subjects in the MSU-MFSD database were divided into two subsets: 15 for training and 20 for testing.

### 4.2. Evaluation Metrics

The comparative results for cross-scenario testing are expressed in terms of the HTER, which is the mean of the false acceptance rate (FAR) and false rejection rate (FRR). On the development set, we first computed the EER and the corresponding threshold, and then used the threshold to determine the HTER on the testing set. Additionally, the receiver operating characteristic (ROC) is reported to assess the method’s performance. We used HTER for the IDIAP Replay-Attack dataset and EER for the CASIA-FASD and MSU-MFSD datasets for intra-scenario testing. We employed area under the curve (AUC) as a performance metric for type-scenario testing.

For a deeper idea about the computational cost of the proposed work, we calculated the overall cost using two different execution environments, CPU and GPU; the obtained results are shown in Table 1. We can observe from Table 1 that the difference between passing one frame as a simple classification model or dealing with the whole video (average 30 frames) is not due much to the use of batches to obtain the results of all the frames, then apply the majority vote on their predicted results. The experiments were conducted using a desktop with an Intel(R) Core.

### 4.3. Performance Comparison on Intra-Database

We computed the EERs for the seven scenarios, including different quality and media, to meet the official CASIA-FAS test protocol. Low, normal, and high-quality image sequences were provided as quality descriptors, and the used media for presentation attack were warped images, chopped photos, and videos played on an iPad. The last scenario was the overall test, which looked at how image quality and spoofing media affected system performance. In this part, we computed two tests, the first of which was performed per-frame and the second of which was performed per-video. The first test used BS-CNN to determine whether an image was real or fake, while the second test used MV to determine whether a video was real or not (See Table 2). In addition, we discovered that the proposed method (BS-CNN+MV) improved the performance.

Moreover, we found that combining MV with BS-CNN yields the best results for picture quality (low, normal, and high), as well as with spoof media (warped photo, cut photo, and video attacks) (see Figure 6). This can be explained by the fact that MV improves decision-making performance. Table 2 shows that our proposed technique outperforms the CASIA baseline in all scenarios when compared to the database created by CASIA [39].

The suggested method is compared to state-of-the-art methods in Table 3 using data from the Replay-Attack database, despite the fact that our EER and HTER are similar to the previous approaches. The following are the types of attacks on replay databases that we computed the performance of using our method: depending on the method used to hold the attack replay device (paper, mobile phone, or tablet), the three attack subsets (print, mobile, and high-def) were recorded in two different modes: (i) fixed-support and (ii) hand-based (See Table 4). We also put our method to the test using the MSU-MFSD database. (See Table 5). It should be noted that no articles have been published that detail the results of various sorts of attacks on this database. We can see that our results are better than the state-of-the-art, with our BS-CNN+MV providing the best results.

The final comparison results on the intra-database are shown in (Table 2, Table 3 and Table 5), which indicates that our proposed method achieves much lower errors on all three datasets than other state-of-the-art methods. Meanwhile, from Figure 7 it can be observed that BS-CNN+MV performs better than BS-CNN, which further verifies the effectiveness of the proposed background subtraction-based convolution neural network.

### 4.4. Inter-Dataset Cross-Type Testing

In this part of the experiments, CASIA-FASD, Replay-Attack, and MSU-MFSD were used to perform the intra-dataset cross-type testing between replay and print attacks. As shown in Table 6, our proposed method outperforms the state-of-the-art methods in terms of overall performance (99.99% AUC), indicating that learned features extended well to unknown attacks. As a result, it appears that our method can learn intrinsic material patterns from a wide range of materials and so generalizes well to previously unexplored types of material.

### 4.5. Inter-Dataset Cross-Dataset Testing

This experiment includes six cross-dataset testing protocols. The first is that we perform CASIA-FASD training and testing on Replay-Attack, which is known as protocol CR; the second is that we perform CASIA-FASD training and testing on MSU-MFSD, which is known as protocol CM; and the third is that we exchange the training and testing datasets that we have in the first, which is known as protocol REPLAY CASIA (RC). The rest of the protocols are identical, with the exception that we utilize the data to train one time and test the next; the protocols are protocol RM, protocol MSU CASIA (MC), and protocol MSU REPLAY (MR), as may be seen in Table 7. Regarding protocol CR, our suggested BS-CNN+MV has 17.62% HTER, exceeding the previous state-of-the-art by a convincing margin of 2%. Increasing the size of the training set with data augmentation could increase performance even more. For protocol RC, we also outperformed state-of-the-art frame-level techniques (see Table 7, third column). Furthermore, we can see in the same table for our suggestion that the convincing margin between protocols RC and CR is 3% in the same technique, compared to the other most convincing methods in the same protocol, such as in [29] (17%). As a result, we can presume that our approach outperforms current approaches.

## 5. Conclusions and Future Directions

Fake face detection is a problem that was addressed in this work. We analyzed seven scenarios from the MSU-MFSD, the REPLAY-ATTACK, and the CASIA-FASD databases. In fact, texture and motion-based characteristics were used by the majority of authors in the field of face presentation attack detection. However, BS and a CNN with a majority vote seems to determine well if a person is using a fake face. In our paper, we evaluated our approach under different protocols. Firstly, we used all three types of databases to evaluate if they produced satisfactory results when compared to the current state-of-the-art. The proposed technique was then put to the test using cross-type testing to ensure that it could handle all attributes and attacks across the three databases. In the final test, we used cross-dataset testing to compare each train of any data with the test, to other data, in order to improve the validity of our approach. The obtained results have shown that our proposed methods outperform the current state-of-the-art. We should point out that the BS-CNN+MV investigation is still in its early stages, as evidenced by the limitations listed below; (1) the limitation of the state-of-the-art face presentation attack detection methods is that they require a perfect result in real-time. Thus, we envision trying to tackle and investigate this problem by using cross-database performance, database limitations, and usability; (2) the problem in FAS with 3D masks or makeup cannot be detected with our proposal. So we need to handle it in future work by using background subtraction with 3D CNN models; (3) applicability to various vision tasks with domain generalization. As a future direction, face presentation attack detection research could focus on making the system more robust across all databases. It is also of interest to create a common training model for each face presentation attack detection using transformers.

## Figures and Tables

**Figure 1 sensors-22-03760-f001:**
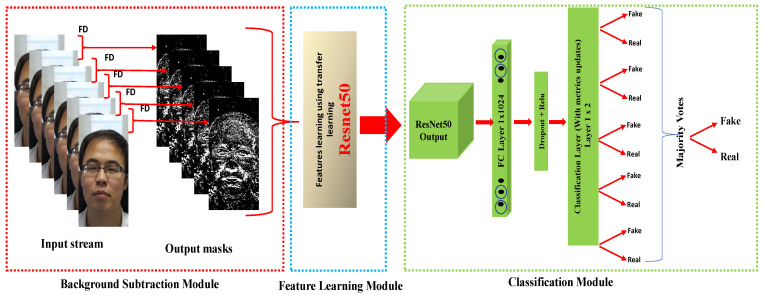
Framework of our proposed approach.

**Figure 2 sensors-22-03760-f002:**
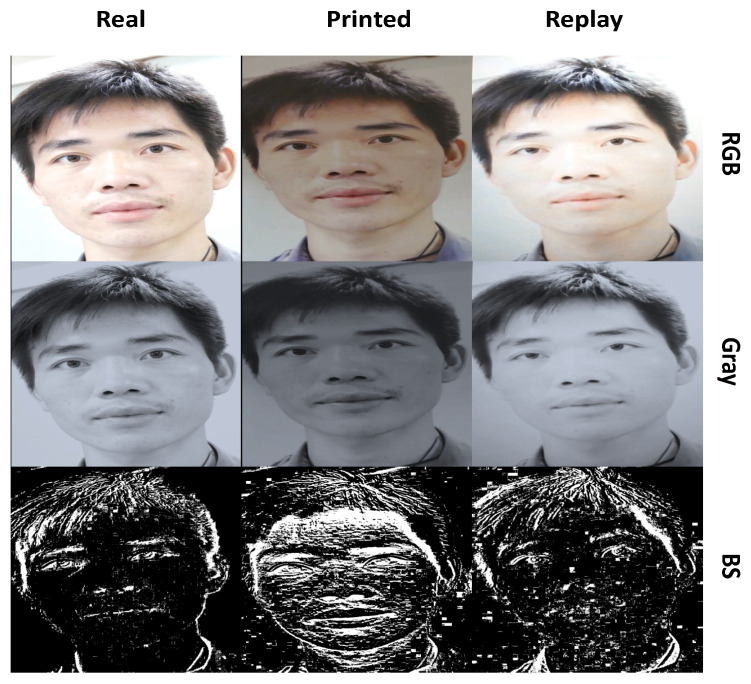
Example of a genuine face and corresponding print and replay attacks in grey-scale and BS.

**Figure 3 sensors-22-03760-f003:**
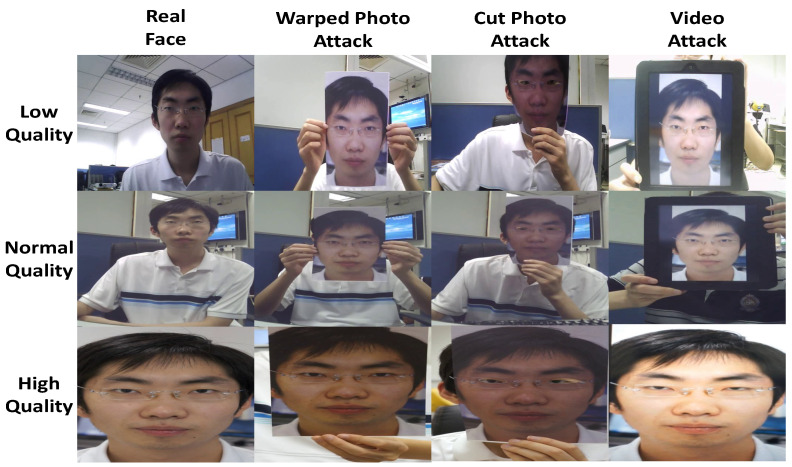
Samples from the CASIA FASD database.

**Figure 4 sensors-22-03760-f004:**
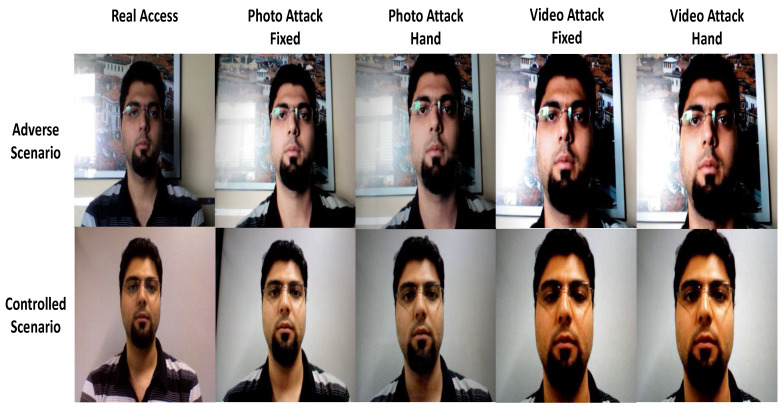
Examples of real accesses and attacks in different scenarios.

**Figure 5 sensors-22-03760-f005:**
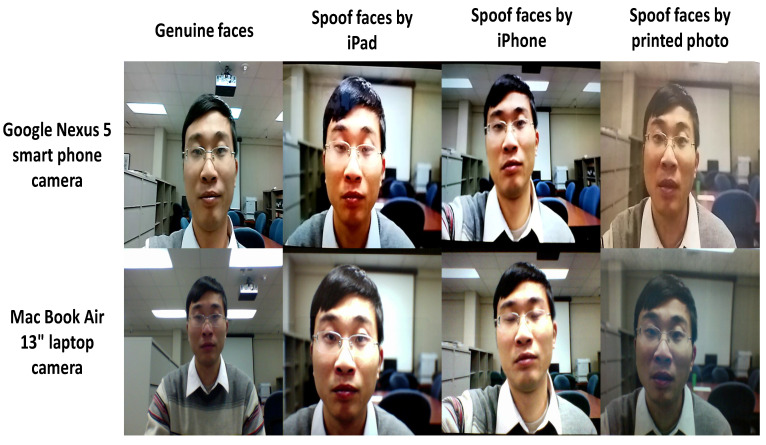
Example images of genuine and attack presentation of one of the subjects in the MSU-MFSD database.

**Figure 6 sensors-22-03760-f006:**
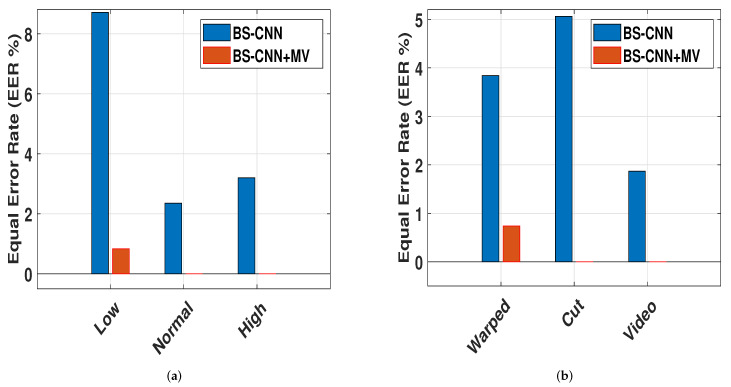
Effect of quality and spoofing media on the performance on the CASIA-FASD. (**a**) Quality and (**b**) spoofing media.

**Figure 7 sensors-22-03760-f007:**
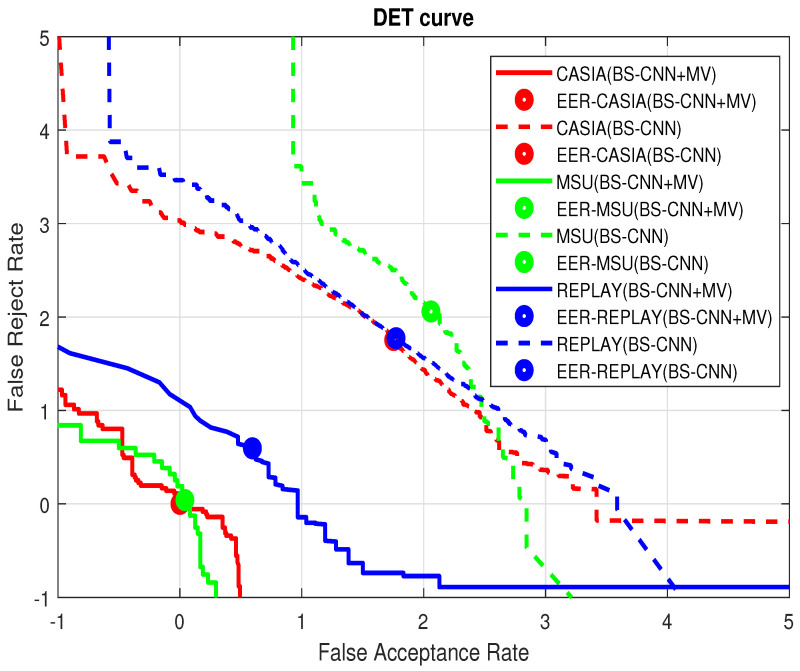
DET curve of the proposed approach on CASIA, MSU, and REPLAY databases.

**Table 1 sensors-22-03760-t001:** Overall computational cost.

		Frame		Video	
		Time	Accuracy	Time	Accuracy
	CPU	2.08 s		3.23 s	
Inference	GPU	0.96 s	95.70 %	2.21 s	99.72 %

**Table 2 sensors-22-03760-t002:** Comparison EER (in %) between the proposed approach and the state-of-the-art methods on different scenarios on CASIA FASD.

	Scenarios						
**Methods**	**Low**	**Normal**	**High**	**Warped**	**Cut**	**Video**	**Overall**
IQA [41]	31.70	22.20	05.60	26.10	18.30	34.40	32.40
DoG baseline [39]	13.00	13.00	26.00	16.00	06.00	24.00	17.00
visual codebooks [42]	10.00	17.78	13.33	07.78	22.22	08.89	14.07
LBP-overlapping+fisher [43]	07.20	08.80	14.40	12.00	10.00	14.70	13.10
CDD [44]	01.50	05.00	02.80	06.40	04.70	00.30	11.80
ML-LPQ fisher [45]	12.49	08.96	05.22	13.62	09.66	10.10	11.39
LBP- TOP [12]	10.00	12.00	13.00	06.00	12.00	10.00	10.00
FD-ML-BSIF-FS [7]	07.93	11.85	12.42	05.85	03.11	15.84	09.96
MLLBP+MLBSIF [46]	06.56	09.93	07.36	09.98	03.45	10.04	09.81
Kernel Fusion [11]	00.70	08.70	13.00	01.40	10.10	04.30	07.20
YCbCr+HSV-LBP [16]	07.80	10.10	06.40	07.50	05.40	08.10	06.20
Identity-DS [22]	-	-	-	-	-	-	03.30
USDAN-Norm [23]	-	-	-	-	-	-	01.10
S-CNN+PL+TC [47]	-	-	-	-	-	-	00.69
BS-CNN+MV (Ours)	00.83	00.00	00.00	00.74	00.00	00.00	00.00

**Table 3 sensors-22-03760-t003:** Comparison between the proposed countermeasure and the state-of-the-art methods on the REPLAY-ATTACK database.

Methods	Overall	
	EER	HTER
IQA [41]	00.00	15.20
LBP [40]	13.90	13.87
MotionCorrelation [48]	11.78	11.79
LBP-TOP [12]	07.90	07.60
IDA [14]	08.58	07.41
Motion+LBP [49]	04.50	05.11
FD-ML-LPQ-Fisher [7]	05.62	04.80
DMD [13]	05.30	03.75
Colour-LBP [16]	00.40	02.90
Spectral cubes [42]	-	02.75
CNN [21]	06.10	02.10
USDAN-Norm [23]	-	00.30
Bottleneck Feature Fusion+NN [24]	00.83	00.00
Identity-DS [22]	00.20	00.00
S-CNN+PL+TC [47]	0.36	-
BS-CNN+MV (our)	00.58	00.62

**Table 4 sensors-22-03760-t004:** Testing our proposed countermeasure using all scenarios of the REPLAY-ATTACK database.

		BS-CNN+MV (Our)	
		EER	HTER
Scenarios	Digitalphoto	01.25	01.87
	Highdef	01.42	03.43
	Mobile	00.00	00.31
	Photo	00.53	02.50
	Print	00.83	00.62
	Video	00.00	01.56
	Overall	00.58	00.62

**Table 5 sensors-22-03760-t005:** Comparison EER (in %) between the proposed approach and the state-of-the-art methods in different scenarios on MSU-MFSD.

	Scenarios						
**Methods**	**HD Android**	**HD Laptop**	**Mobile Android**	**Mobile Laptop**	**Print Android**	**Print Laptop**	**Overall**
IDA [14]	-	-	-	-	-	-	08.58
Identity-DS [22]	-	-	-	-	-	-	08.58
FD-ML-BSIF-FS [7]	-	-	-	-	-	-	02.10
S-CNN+PL+TC [47]	-	-	-	-	-	-	00.64
USDAN-Norm [23]	-	-	-	-	-	-	00.00
BS-CNN+MV (our)	00.00	00.00	00.00	00.00	00.00	00.00	00.00

**Table 6 sensors-22-03760-t006:** AUC (%) of the model cross-type testing on CASIA-FASD, Replay-Attack, and MSU-MFSD.

Methods	CASIA-FASD			Replay-Attack			MSU-MFSD			Overall
	Video	Cut Photo	Wrapped	Video	Digital Photo	Printed	Printed	HR Video	Mobile Video	
OC-SVM+BSIF [50]	70.74	60.73	95.90	84.03	88.14	73.66	64.81	87.44	74.69	78.68±11.74
NN+LBP [51]	94.16	88.39	79.85	99.75	95.17	78.86	50.57	99.93	93.54	86.69±16.25
SVM+LBP [52]	91.94	91.70	84.47	99.08	98.17	87.28	47.68	99.50	97.61	88.55±16.25
NAS-Baseline [26]	96.32	94.86	98.60	99.46	98.34	92.78	68.31	99.89	96.76	93.90±09.87
DTN [25]	90.00	97.30	97.50	99.90	99.90	99.60	81.60	99.90	97.50	95.90±06.20
AIM-FAS [27]	93.6	99.7	99.1	99.8	99.9	99.8	76.3	99.9	99.1	96.40±07.80
CDCN [29]	98.48	99.90	99.80	100.00	99.43	99.92	70.82	100.00	99.99	96.48±09.64
CDCN++ [29]	98.07	99.90	99.60	99.98	99.89	99.98	72.29	100.00	99.98	96.63±09.15
BCN [28]	99.62	100.00	100.00	99.99	99.74	99.91	71.64	100.00	99.99	96.77±09.99
NAS-FAS [26]	99.62	100	100	99.99	99.89	99.98	74.62	100.00	99.98	97.12±08.94
BS-CNN+MV (our)	100	100	99.98	100	100	100	100	100	100	99.99±0.0067

**Table 7 sensors-22-03760-t007:** The results of cross-dataset testing among CASIA-FASD, MSU-MFSD, and Replay-Attack. The evaluation metric is HTER(%).

Method	Protocol CR		Protocol CM		Protocol RC		Protocol RM		Protocol MC		Protocol MR	
	Training	Testing	Training	Testing	Training	Testing	Training	Testing	Training	Testing	Training	Testing
	Casia	Replay	Casia	MSU	Replay	Casia	Replay	MSU	MSU	Casia	MSU	Replay
FD-ML-LPQ-FS [7]	50.25		50.41		42.59		38.00		50.00		48.00	
Motion-Mag [53]	50.10		NP		47.00		NP		NP		NP	
LBP-TOP [54]	49.70		NP		60.60		NP		NP		NP	
LBP [16]	47.00		NP		39.60		NP		NP		NP	
Spectral cubes [42]	34.40		NP		50.00		NP		NP		NP	
STASN [30]	31.50		NP		30.90		NP		NP		NP	
Color Texture [55]	30.30		NP		37.70		NP		NP		NP	
FaceDs [33]	28.50		NP		41.10		NP		NP		NP	
Auxiliary [31]	27.60		NP		28.40		NP		NP		NP	
MEGC [56]	20.20		NP		27.90		NP		NP		NP	
FAS-TD [32]	17.50		NP		24.00		NP		NP		NP	
BASN [34]	17.50		NP		24.00		NP		NP		NP	
Patch+BCN+MFRM [28]	16.60		NP		36.40		NP		NP		NP	
CDCN [29]	15.50		NP		32.60		NP		NP		NP	
BS-CNN+MV (our)	17.62		23.75		20.35		24.16		35.45		44.33	

## Data Availability

Not applicable.

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
