# Peer review of "Face Presentation Attack Detection Using Deep Background Subtraction"

_sensors, 2022, doi:10.3390/s22103760_

Round 1

Reviewer 1 Report

This paper is about face spoof attack detection using deep background
subtraction. I have the following comments:

  1. Remove the period before [5]
  2. At the end of related work section, please add the limitations of related work. This will be the introduction to your contributions
  3. What is the rationale for using RESNET-50 and not other transfer learning model?
  4. Deep learning models such as CNN, can perform both feature extraction and classification at the the same time. The proposed model uses RESNET for feature extraction then cascaded with classification. This would increase the complexity of the system. So a fair comparison should be conducted between the proposed model and the CNN model in terms of accuracy and complexity
  5. How the hyperparameters of the deep learning model are tuned? have you used any optimization algorithms.
  6. In Table 2, the EER of the proposed model is almost zero. So studying the complexity of the model is very essential
  7. Add a paragraph to describe the limitations of the system

Author Response

First of all, we would like to thank the AE for handling our article and the reviewer for their valuable comments and suggestions.  We have taken into account all the suggestions of the reviewer.  We provide below our detailed answers to the reviewer remarks and highlight the implemented revisions in the paper.  We hope that this revised version fulfills all the suggestions of the reviewers. More precisely, we have done the following:

Reviewer 2 Report

The authors present a new approach for face spoof detection based on background subtraction, deep neural networks and ensemble voting. Given a video sequence, background subtraction between consecutive frames extracts motion. Next, feature extraction is performed on each frame by using ResNet50. Finally, the extracted features of each frame are classified as fake or real. The classification scores of each frame are combined by using majority voting to obtain the predicted class.

The approach has been tested on several datasets: CASIA-FASD, Replay-Attack, MSU-MFSD. The results show that the proposed approach outperforms state-of-the-art approaches.

The paper needs a minor spell check. Some typos and corrections can be found in the attachment (sensors-1688893-peer-review-v1-reviewed.pdf).

Author Response

(The authors gave the same response as above.)

Reviewer 3 Report

The authors present a face presentation attack detection (PAD) method based on background subtraction and fine-tuning of a ResNet-50 CNN to derive a majority vote of all frames of the input video. This algorithm is evaluated on three datasets containing only printout and screen replay attacks. This brings me already to the major drawback of this work. These attacks are kind of easy to detect in contrast to more sophisticated attacks like masks or makeup. Hence, it is already state-of-the-art that printouts and screen replays are no real threat to face recognition systems. Hence, this article needs a major revision including new experiments on different datasets to achieve the novelty worth publishing in a journal. Since, the Sensors timeframe only allows one week for revision, which is not enough time to run new experiments, I vote to reject this article. Please find below additional comments how to improve the quality of article.

1) I would like to introduce ISO/IEC 30107-1 [1], which standardised terms and definitions for presentation attack detection. Please update the manuscript according to these definitions:
anti-spoofing -> presentation attack detection
spoof attack -> presentation attack instrument
face spoofing -> face presentation attack detection
live face -> bona fide presentation / bona fide face
spoof face -> attack presentation

2) In the introduction you state that there are only two types of attacks: print and replay. This is simply wrong! You can also attack face recognition systems using e.g. silicon masks, makeup, etc. [2]. In fact, the database in [2] is freely available, so you can additionally evaluate your PAD method on more sophisticated attacks.

3) It remains unsure whether you train and test your PAD method on combined data from all used datasets or whether you train and test three separate models. Additionally, please add the information on how many frames of the video are used. It is a difference if the video length is 30 frames or 300 frames.

4) The used evaluation metrics are defined to measure the biometric systems recognition performance [3]. The metrics for PAD are defined in ISO/IEC 30107-3 [4] as:
* Attack presentation classification error rate (APCER): the proportion of attack presentations falsely classified as bona fide presentations.
* Bona fide presentation classification error rate (BPCER): the proportion of bona fide presentations falsely classified as attack presentations
* Detection Equal Error Rate (D-EER): point where APCER = BPCER
Please adjust the manuscript accordingly.

i) The sensors template includes a table for abbreviations, however they should be at the end of the article before the references. Please move table 1 accordingly, it does not belong on page 2.

ii) Please introduce abbreviations in the text once and do not rely only on table 1. Please note that the abstract does not count as text. On the other hand, you do not need to introduce abbreviations of related work, which you do not use again.

iii) Figure 5 is mentioned earlier than figure 4. Either swap the figures or the texts. Also table 4 is mentioned earlier than table 3.

iv) Figure 7 does not really look like a DET. Maybe you can double check your code with the official NIST sources [5].

------
[1] ISO/IEC JTC1 SC37 Biometrics. ISO/IEC 30107-1. Information Technology - Biometric Presentation Attack Detection - Part 1: Framework, 2016.

[2] Mostaani, Z., George, A., Heusch, G., Geissbuhler, D., & Marcel, S. (2020). The high-quality wide multi-channel attack (HQ-WMCA) database. arXiv preprint arXiv:2009.09703.
Download: https://www.idiap.ch/dataset/hq-wmca

[3] ISO/IEC JTC1 SC37 Biometrics, ISO/IEC 19795-1:2021. Information Technology – Biometric Performance Testing and Reporting – Part 1: Principles and Framework, ISO/IEC, 2021.

[4] ISO/IEC JTC1 SC37 Biometrics. ISO/IEC 30107-3. Information Technology - Biometric Presentation Attack Detection - Part 3: Testing and Reporting, 2017.

[5] DETware can be downloaded from https://www.nist.gov/itl/iad/mig/tools

Author Response

(The authors gave the same response as above.)

Reviewer 4 Report

The paper is devoted to many aspects that are associated with the detection of a face spoof attack detection. The authors of the paper describe in detail the main problems that arise in the process of solving problems related to automatic detection when someone tries to deceive a system using a face spoof attack detection. The paper clearly shows that despite the great practical potential, the problem of effective recognition of a face spoof attack detection has not yet been solved. Therefore, it can be said that currently operating fully automated models and methods capable of 100% detecting of a face spoof attack detection in biometric systems do not currently exist in a full-fledged form. To create such full-fledged models, it is necessary to perform a deep analysis of visual signs that could clearly determine when a person’s face is not reliable. The authors of the paper also describe in detail the various transfer learning methodologies and highlight their main positive and negative points (especially in tasks related to face forgery). Testing and comparative analysis of the results of the study was carried out on a variety of data sets. Comparative analysis showed that qualitative indicators still depend on the input data, but on average demonstrate high performance. The provisions of the paper and the main conclusions are reasoned, correspond to the current level of development of the chosen field of study. However, the article is not free from a number of shortcomings. First of all, the question arises why ResNet-50 was chosen as the neural network model? I would like to see the results of fine-tuning and other neural network models in order to be sure that this particular fine-tuning is better than others. The question also arises, will the results be better if the data method adds a stage with MixUp and, for example, an attention module (Squeeze-and-Attention), as well as smoothing (Label Smoothing)? It seems to me that all the proposed additions will only improve this paper, and it will be useful and interesting to many. Finally, the style of the paper requires minor revision due to the presence of spelling and punctuation errors.
In general, the paper is interesting, but it should be finalized and only then it can be accepted for publication.

Author Response

(The authors gave the same response as above.)

Round 2

Reviewer 1 Report

The authors addressed my comments

Reviewer 3 Report

The article has no significant contribution to the current state-of-the-art since the experiments are only evaluating printout and screen replay attacks. I see that other points were addressed in the revision but this is not enough for publishing the manuscript.